# Role of Cytokines in EGPA and the Possibility of Treatment with an Anti-IL-5 Antibody

**DOI:** 10.3390/jcm9123890

**Published:** 2020-11-30

**Authors:** Takeo Isozaki, Tetsuya Homma, Hironori Sagara, Tsuyoshi Kasama

**Affiliations:** 1Division of Rheumatology, Department of Medicine, Showa University School of Medicine, Tokyo 142-8666, Japan; tkasama@med.showa-u.ac.jp; 2Division of Respiratory Medicine and Allergology, Department of Medicine, Showa University School of Medicine, Tokyo 142-8666, Japan; oldham726@yahoo.co.jp (T.H.); sagarah@med.showa-u.ac.jp (H.S.)

**Keywords:** EGPA, cytokines, IL-5, mepolizumab

## Abstract

Eosinophilic granulomatosis with polyangiitis (EGPA) is a type of systemic vasculitis with eosinophilia in the peripheral blood, which is preceded by bronchial asthma or allergic disease. EGPA is pathologically characterized by microangiopathy granulomatosis vasculitis. Vasculitis can be exacerbated and cause central nervous system and cardiovascular disorders and gastrointestinal perforation. Histological examination reveals eosinophil infiltration and granulomas in lesions in areas such as the lung, nervous system, and skin. Laboratory tests show inflammatory findings such as C-reactive protein (CRP) elevation, increased eosinophils, elevated serum IgE, and elevated myeloperoxidase-anti-neutrophil cytoplasmic antibodies (MPO-ANCA). MPO-ANCA is positive in approximately 40–70% of cases of this disease. EGPA is a necrotizing vasculitis that affects small- and medium-sized blood vessels; however, it differs from other types of ANCA-related vasculitis (such as microscopic polyangiitis and granulomatosis) because it is preceded by bronchial asthma and eosinophilia in the blood and tissues. Treatment with immunosuppressive agents such as steroids or cyclophosphamide depends on the Five Factor Score, which predicts the prognosis and severity of the condition. If the effect of appropriate treatment with steroids is insufficient, the anti-interleukin-5 antibody mepolizumab can be administered. The combination of mepolizumab with standard treatment leads to a significantly longer duration of remission, a higher proportion of patients who achieve sustained remission, and less steroid use than with a placebo.

## 1. Introduction

Eosinophilic granulomatosis with polyangiitis (EGPA) was described in 1951 by J. Churg and L. Strauss as a form of disseminated necrotizing vasculitis with extravascular granulomas that occurred in patients with asthma and tissue eosinophilia [1]. EGPA is pathologically a granulomatous inflammation of the respiratory tract, often with eosinophil infiltration and necrotizing vasculitis in small- and medium-sized vessels [2,3]. Asthma is present in 96% to 100% of EGPA patients and is a major feature of EGPA. The involvement of type 1 allergy has been suggested due to the spread of allergic disease. When an allergic patient is stimulated by antigens from the respiratory tract, airway mucosa, mast cells, macrophages, T cells, and eosinophils produce eotaxin, which mobilizes eosinophils and cytokines and activates eosinophils [4,5,6]. Eosinophils secrete major basic proteins that damage tissues from eosinophil granules, eosinophil peroxidase, and platelet-activating factors, which are involved in the exacerbation of bronchial asthma and lesions, leading to peripheral neuropathy and myocardial damage [7,8,9]. In addition, inflammatory cytokines such as tumor necrosis factor-α, interleukin (IL)-1β, and IL-8/CXCL8 are produced in response to antigen stimulation, and vascular endothelial cell damage due to degranulation and immune complex deposition associated with neutrophil activation leads to necrotizing vasculitis [10,11]. Activated T cells produce macrophage chemotactic factors, macrophage activating factors, and IL-5, which activate macrophages and cause granuloma formation. Granuloma formation involves the influx and accumulation of phagocytic monocytes in vascular lesions, aggregation and organization of embryonic monocytes and mature macrophages, and eventually their development into epithelioid cells. IL-5 is also involved in eosinophil recruitment [12,13,14].

## 2. EGPA Diagnostic Criteria

The American College of Rheumatology (ACR) 1990 criteria are often used as diagnostic criteria. According to the ACR classification criteria, satisfying four or more out of six items can detect this disorder with a high sensitivity of 85.0% and specificity of 99.7%, and these criteria are easily applied in clinical settings [15] (Table 1).

## 3. Damage to Each Organ Due to EGPA

This disease is characterized by a combination of vasculitis in organs throughout the body and organ damage associated with eosinophil infiltration. The damaged organs include the joints, skin, lung, myocardium, peripheral nerves, and gastrointestinal lesions.

EGPA usually develops in adulthood and subjects suffer from severe asthmatic and eosinophilic sinusitis symptoms that arise from prominent eosinophilic hyperplasia [16,17]. The severe type of eosinophilic sinusitis is often associated with nasal polyps, which cause decreased olfaction, which impairs quality of life (QOL). This nasal polyp complication is useful for differentiating EGPA from hypereosinophilic syndrome (HES). In addition, 30% of subjects suffer from a precursive chronic eosinophilic pneumonia. A recent report by Berti et al. showed that uncontrolled asthmatic symptoms are associated with baseline pulmonary, ear, nose, and throat manifestations, but not with vasculitic symptoms among asthmatic patients with EGPA [17]. An atopic factor was also related to a prognosis among asthmatic patients with EGPA [18]. These lines of evidence suggest that controlling the airway symptoms may alter their courses and QOL.

Other than vasculitis symptoms, such as fever, myalgia, and rapid weight loss, symptoms from multiple mononerve inflammations (numbness of peripheral limbs and muscle weakness) are commonly observed in 90% or more of the cases. Numbness in the limbs, symptoms of not being able to hold chopsticks, and foot drops appear. In addition, about half of the cases are accompanied by skin symptoms (purple spots, etc.), heart disorders (palpitations, arrhythmia, heart failure symptoms, etc.), and gastrointestinal ischemic symptoms (abdominal pain, vomiting, diarrhea, gastrointestinal bleeding due to ileus, intestinal ulcer, etc.). Blood tests show anemia, eosinophilia, and increased C-reactive protein (CRP), lactate dehydrogenase (LDH), and creatine kinase (CK,) which are indicators of ischemia of various organs. The degree of eosinophilia varies, but it is often 30% or more, observed at the onset of vasculitis and is accompanied by leukocytosis. However, in rare cases, eosinophilia is not noticeable when systemic steroid therapy is already used in combination. In two-thirds of the cases, a marked increase in total serum IgE level, rheumatoid factor positivity, and an increase in platelet count are observed. The positivity rate of MPO-ANCA (P-ANCA), which has been emphasized as an anti-neutrophil antibody in EGPA, has not been at a high level in recent years, and is only present in 30% to 40% of the cases. On the other hand, unlike GPA, PR3-ANCA (C-ANCA) is positive in less than 10% of the cases.

## 4. The Role of Cytokines and Chemokines

Various types of cells are involved in the formation of the pathophysiology of allergic diseases [19]. In bronchial asthma, inflammatory cells such as lymphocytes, mast cells, and eosinophils infiltrate the lesion site due to allergen stimulation, and bronchial epithelial cells, fibroblasts, and smooth muscle cells interact with each other to produce mucus. It forms pathological conditions such as thickening of the submucosa and hyperplasia of smooth muscle. Cytokines and chemokines play a role in mediating the actions between these various cells. Helper T cells are classified into Th1 type and Th2 type according to the pattern of secreted cytokines or chemokine receptors expressed on the cell surface. In the active phase of EGPA, not only type 2 cytokines, such as IL-4 and IL-5, but also IL-10, which is a suppressive cytokine, are present. The production of inflammatory cytokines such as tumor necrosis factor (TNF)-α and interferon-γ is increased [20,21]. According to other findings, the soluble interleukin-2 receptor (sIL-2R) is a serum marker for EGPA, and eosinophilic cationic protein levels are upregulated and correlated with the eosinophil blood count in this disease [22]. Various biomarkers for EGPA have been investigated; however, no useful biomarkers have been identified to date. Kiene, F et al. reported significant increases in IL-4 and IL-13 production in T cells from patients with EGPA compared with that in patients with another granulomatous disease or healthy controls. In addition, the positive correlation between IL-4 production and the eosinophil count in EGPA, in contrast to the more Th1-associated cytokine pattern found in granulomatous disease, supports our assumption that EGPA is a Th2-associated disease [23]. In a study of bronchoalveolar lavage fluid (BALF) in EGPA and bronchial asthma patients, EGPA had a Th2 response compared to bronchial asthma, and clinical parameters of disease activity were strongly correlated with the expression of IL-4, IL-5, IL-10, and STAT5A [20]. However, these biomarkers are not specific to EGPA because they are all elevated in diseases that cause strong eosinophilic inflammation [11]. In addition, the ratio of the peripheral blood eosinophil count, IgE, CRP and TARC/CCL17, eotaxin-3/CCL26, and IgG4 have been investigated. None of these indicated activity and inactivity [6,24]. IL-5 is increased in active EGPA, and its inhibition could represent a potential therapeutic target [25,26]. This section is divided by subheadings. It should provide a concise and precise description of the experimental results, their interpretation, as well as the experimental conclusions that can be drawn.

### 4.1. Biomarkers of Asthma/Eosinophilic Disorders

#### 4.1.1. IL-5

IL-5 is a cytokine that belongs to the β common chain family and binds a heterodimer receptor, IL-5R, and common β subunit βc [27]. IL-5 and IL-5R are involved in maintaining the survival of eosinophils, developing allergic inflammation, and maintaining the pathology [28,29]. It has become clear that IL-5 is constitutively produced not only by Th2 but also by innate lymphoid cells, and constantly regulates innate immunity and natural inflammation. Group 2 innate lymphoid cells (ILCs) are an important source of interleukin-5, contributing to tissue and blood eosinophilia [30]. IL-25 stimulates Th2 cells and group 2 ILCs to markedly increase the production of IL-5 [31]. Based on reports that human IL-5 is involved in the pathophysiology of eosinophilic inflammation, eosinophilia and eosinophilic inflammation were used with humanized anti-IL-5 antibodies and anti-human IL-5Rα chain antibodies. Exposure of the allergen to the bronchial mucosa of patients with bronchial asthma results in a biphasic response consisting of immediate bronchoconstriction and delayed airway obstruction. Furthermore, eosinophil infiltration into the alveoli is induced and airway hyperresponsiveness is enhanced. Targeting IL-5 or IL-5Rα is an attractive approach to the treatment of patients with eosinophilic bronchial asthma. The anti-IL-5 monoclonal antibody binds to IL-5 and interferes with ligation to IL-5Rα expressed on eosinophils and basophil membranes [32].

#### 4.1.2. IL-4

IL-4 is also a Th2 cytokine that regulates multiple biological functions [33]. The effect of IL-4 signaling is mediated through the IL-4 receptor α chain (IL-4Rα). By binding to IL-4Rα, it dimers with a common gamma chain (γc) to produce a type 1 signaling complex located primarily in hematopoietic cells [34]. When the IL-4 signal is activated, the type 1 complex transmits the signal via the Janus family kinases (JAK1 and JAK3), phosphorylates the transcription factor STAT6, and then dimerizes into the cell nucleus. STAT6 promotes transcription of GATA3 (Th2 cell inducer) and MHCII (myeloid and B cells) and induces IgE class switching in B cells [35,36]. JAK1 also phosphorylates insulin receptor substrates −1 and −2, and are activated and promote survival and growth via phosphoinositide 3 (PI3)/AKT, protein kinase B (PKB)/mTOR, and other pathways [37]. Patients with bronchial asthma have elevated levels of IL-4 protein in serum and bronchoalvelar lavage fluid (BALF), increased IL-4 mRNA and protein on bronchial biopsies, and express IL-4 mRNA in BALF and bronchial biopsies [38,39,40,41]. In mice, neutralizing IL-4 with anti-IL-4 antibodies suppresses the development of allergen-specific IgE and reduces eosinophilic inflammation and airway responsiveness. IL-4-deficient mice maintain a residual Th2 response. This may explain persistent low levels of IL-5 expression, eosinophilia, and airway hyperresponsiveness observed in some of these mouse studies [42].

#### 4.1.3. IL-13

IL-13 is a prominent Th2 cytokine released from Th2 cells. The effect of IL-13 on immune cells is similar to that of IL-4 because these cytokines share a common receptor subunit (α subunit), but IL-13 is associated with bronchial hypersensitivity. It plays a major role in allergic inflammation by promoting the overproduction of mucus. In a mouse model, IL-13 is an important mediator of allergic asthma and induces IL-5 and eotaxin/CCL11 dependent eosinophil recruitment into the airways. IL-13 has almost the same effect on human B cells as IL-4, but the effect is generally less potent. No additive or synergistic effects are observed when both cytokines are added at optimal concentrations. IL-13 contributes to the allergic inflammatory process through its ability to induce the expression of vascular cell adhesion molecules (VCAMs) −1 in human umbilical vein endothelial cells [43]. Overexpression of IL-13 in the lungs of mice caused hypersecretion of mucus, subepithelial fibrosis, eotaxin production, and eosinophil infiltration [44]. Reconstitution of STAT6 in epithelial cells was only sufficient for IL-13-induced airway hypersensitivity and mucus production in the absence of inflammation and fibrosis. Elevated IL-13 levels are found in the blood of asthmatics, sputum, bronchial mucosa, and BALF compared to healthy controls [45,46,47]. IL-13 (+) ILC2 increased in the circulation of asthma patients and was at a level that correlated with the severity of asthma [48].

#### 4.1.4. IL-10

IL-10 can down-regulate cytokine production from Th2 cells as well as Th1 cells [49,50]. Patients with asthma had a relatively reduced ability to produce IL-10 by BALF and mononuclear cells. These associations led to the speculation that the constitutive expression of IL-10 in the airways may contribute to maintaining a normal state of allergen-free response [51]. However, the role of IL-10 in the regulation of Th2-mediated diseases such as asthma is controversial. Pleiotropic cytokines IL-10 and TGF-β, which have significant anti-inflammatory and immunosuppressive properties, are important regulators in maintaining immunological homeostasis. Relative underproduction of IL-10 in alveolar macrophages and sputum in asthmatics has been reported, suggesting an important role of IL-10 in the regulation of airway inflammation [51,52].

#### 4.1.5. TARC/CCL17

Thymus and activation-regulating chemokine/chemokine ligand 17 (TARC/CCL17) is a member of the CC chemokine group [53]. It is a ligand for the CC chemokine receptor, which is selectively expressed in Th2 cells and helps mobilize and migrate cells with this receptor [54,55]. TARC/CCL17 has been proposed as a candidate gene for conferring susceptibility to Th2 associated with allergic diseases and is deeply involved in the etiology of atopic dermatitis and bronchial asthma [56,57,58]. Similar to CCR4 antibodies, specialized antibodies against TARC/CCL17 and CCL22 can reduce ovalbumin (OVA)-induced airway eosinophilia and hypersensitivity reactions in asthmatic mice [59]. CCR4 and its ligands (CCL17 and CCL22) play important roles in asthma inflammation.

### 4.2. Biomarkers of Vasculitis

Along with eosinophil injury as EGPA, it also has an aspect of vasculitis. Pagnoux, C. et al. showed that macrophage-derived chemokine (MDC)/CCL22, IL-8/CXCL8, MIP-1α, MIP-1β, and TNF-α in EGPA serum were lower than in healthy controls, asthma, or primary hypereosinophilic syndrome [60].

#### Periostin

Periostin is one of the extracellular matrix proteins, has four fasciclin-1 domains in the molecule, binds to cells via integrin molecules, and binds to other extracellular matrix proteins such as fibronectin, tenascin C, and collagen V. In addition to bronchial asthma, it has been reported to be involved in fracture repair, tumor infiltration, and myocardial infarction. Periostin is one of the proteins secreted by epithelial cells in response to stimulation with IL-4 or IL-13 and facilitates eosinophil-mediated type 2 inflammation and fibrosis. Serum periostin is associated with persistent eosinophilic airway inflammation in bronchial asthma [61]. Rhee RL et al. also showed periostin levels in EGPA were significantly higher than in healthy controls and patients with bronchial asthma [62].

## 5. Treatment of EGPA

Steroids are the first-line treatment for EGPA. Severe or rapidly progressing cases are treated with steroid pulse therapy. In severe cases with poor responses to steroid treatment, immunosuppressive drugs are used in combination treatment. Cyclophosphamide is recommended as a combination treatment, especially in patients with a poor prognosis. A positive score of two or more is considered to have a poor prognosis (1. age > 65 years, 2. cardiac insufficiency, 3. renal insufficiency (creatinine > 1.7 mg/dL), 4. gastrointestinal involvement, and 5. the absence of ear, nose, and throat manifestations) [63] (Table 2). Regarding cyclophosphamide therapy, there was no significant difference between 6 and 12 months of treatment for EGPA with poor prognostic factors. The recurrence rate was lower in the group treated for 12 months [64]. High-dose gamma globulin therapy is expected to be effective for peripheral neuropathy and cardiac dysfunction, which are difficult to treat with systemic steroids. Several case reports demonstrated the efficacy of high-dose intravenous immunoglobulin (IVIg) (i.e., 2 g/kg for 2–5-day cycles, which can be repeated every 3–4 weeks) in naïve and previously treated EGPA patients [65,66].

In addition to these steroids and immunosuppressants, treatments targeting cytokines have been established. Mepolizumab and reslizumab are both humanized monoclonal antibodies that bind to and block the function of circulating IL-5 and consequently prevent the binding of IL-5 to its receptor. Treatment with mepolizumab resulted in significantly more weeks of remission than with placebo (28% vs. 3% of patients experienced remission for 24 weeks or longer; odds ratio 5.91; 95% confidence interval (CI), 2.68–13.03; *p* < 0.001) and a significantly higher proportion of those participants remained in remission at 36 and 48 weeks than with placebo (32% vs. 3%; odds ratio 16.74; 95% CI 3.61 to 77.56; <0.001). Forty-four percent of subjects treated with mepolizumab were able to taper off prednisolone or prednisone to less than 4 mg per day, compared with 7% of subjects who received the placebo. The proportion of patients with a time to initial recurrence of over 52 weeks was higher with mepolizumab than with placebo (56% vs. 82%; hazard ratio 0.32; 95% CI 0.21 to 0.50; *p* < 0.001). Adverse events were headache (32% in the mepolizumab group, 18% in the placebo group), nasopharyngitis (18% vs. 24%), arthralgia (22% vs. 18%), sinusitis (21% vs. 16%), upper respiratory tract infection (21% vs. 16%), exacerbation of asthma (3% vs. 6%), and local injection reaction (similar in the two groups) [67].

Kim S et al. reported that there was a significantly lower exacerbation rate during the treatment period (0.14 events per week, two events during a 14-week period) compared with the nontreatment period (0.69 events per week, 18 events over a 26-week period) in EGPA. They also showed mepolizumab effectively served as a corticosteroid-sparing therapy. The mean dose at baseline was 12.9 mg/day, which was reduced to 4.6 mg/day after 12 weeks of therapy, that is a 64% reduction in the corticosteroid dose after mepolizumab therapy [68]. There are other reports investigating the effects of corticosteroid dosage. Moosing et al. showed that the daily dose of glucocorticoid was reduced significantly at week 32 (median, 19 mg at baseline to 4 mg at week 32; *p* = 0.006) [26]. On the safety side, mepolizumab was well tolerated and the most common adverse events associated with mepolizumab therapy were eczema, edema, swelling of the left hand, urinary tract infection, dentalgia, abdominal pain, wound infection, otitis media, bronchitis, herpes zoster, and herpes simplex. Severe adverse events like anaphylaxis (*n* = 1), norovirus infection (*n* = 1), cerebral micro embolism (*n* = 1), and de Quervain thyroiditis (*n* = 1) were noted. However, these serious adverse events were probably unrelated to mepolizumab therapy [26,69].

There are reports of cases of EGPA in which rituximab, another treatment option, was effective against treatment resistance to steroids and immunosuppressants [70,71]. A prospective study showed that rituximab was superior to azathioprine as a maintenance therapy [72].

## 6. Conclusions

There are diseases similar to EGPA, and there are many situations in which EGPA is difficult to diagnose. Due to the fatal complications, early and appropriate treatment is important. Mepolizumab is effective against eosinophilic bronchial asthma and holds promise for EGPA [73,74]. These drugs have the potential to be useful therapeutic agents.

## Figures and Tables

**Table 1 jcm-09-03890-t001:** Criteria and definitions used for the classification of eosinophilic granulomatosis with polyangiitis (EGPA) (EGPA was the name of Churg-Strauss syndrome in 1990).

Criterion	Definition
Asthma	History of wheezing or diffuse high-pitched rales on expiration
Eosinophilia	Eosinophilia > 10% in the white blood cell differential count
History of allergy	History of seasonal allergies (e.g., allergic rhinitis) or other documented allergies, including to food, contactants, and others (except for drug allergies)
Mononeuropathy or polyneuropathy	Development of mononeuropathy, multiple mononeuropathies, or polyneuropathy (i.e., glove/stocking distribution) attributable to systemic vasculitis
Pulmonary infiltrates, non-fixed	Migratory or transitory pulmonary infiltrates on radiographs (not including fixed infiltrates) attributable to systemic vasculitis
Paranasal sinus abnormality	History of acute or chronic paranasal sinus pain or tenderness or radiographic opacification of the paranasal sinuses
Extravascular eosinophils	Biopsy, including of an artery, arteriole, or venule, showing the accumulation of eosinophils in extravascular areas

History of allergy, other than asthma or drug-related, is included only in the tree classification criteria set and not in the traditional format criteria set, which requires 4 or more of the 6 other items listed here.

**Table 2 jcm-09-03890-t002:** Prognostic criteria predicting survival in EGPA.

Revised 2011 Five-Factor Score
1. Age > 65 years
2. Cardiac insufficiency
3. Renal insufficiency (creatinine > 1.7 mg/dL)
4. Gastrointestinal involvement
5. Absence of ear, nose and throat manifestations

The prognosis is considered poor with 2 or more of the 5 items.

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
