# Peer review of "Role of Cytokines in EGPA and the Possibility of Treatment with an Anti-IL-5 Antibody"

_jcm, 2020, doi:10.3390/jcm9123890_

Round 1
Reviewer 1 Report
I read with interest the article of Isozaki et al on cytokines in EGPA. I would suggest to adress the following minor points, to improve the quality of the paper:
- MPO-ANCA is positive in 10-40% not, 40-70% . Please change- (Sinico et al. Prevalence and clinical significance of antineutrophil cytoplasmic antibodies in Churg-Strauss syndrome. Arthritis Rheum. 2005)
- In the chapter called “damage to each organ” , when you speak of severe asthma please discuss asthma severity at the diagnosis, which is linked to atopy (Berti A. et al. Severe/uncontrolled asthma and overall survival in atopic patients with eosinophilic granulomatosis with polyangiitis.
Respir Med. 2018; 142: 66-72) and long term asthma, which is linked with several factors (Berti A et al. Eosinophilic Granulomatosis With Polyangiitis Clinical Predictors of Long-term Asthma Severity, Chest 2020)
- I would change the structure of the paragraph, trying to divide biomarkers by area, i.e. biomarkers of asthma/eosinophilic disorders (e.g. IL5, Il13) and biomarkers of vasculitis, this last section is lacking. In this section authors should focus on those more inflammatory cytokines. I suggest to cite the paper of Pagnoux. C. et al. Serum cytokine and chemokine levels in patients with eosinophilic granulomatosis with polyangiitis, hypereosinophilic syndrome, or eosinophilic asthma Clin. Exp Rheum. 2019.
- I would add a section on periostin and cite Rhee R et al. Serum periostin as a biomarker in eosinophilic granulomatosis with polyangiitis. PLoS One . 2018 Oct 11;13(10)
Author Response
- MPO-ANCA is positive in 10-40% not 40-70%. Please change. (Sinico et al. Prevalence and clinical significance of antineutrophil cytoplasmic antibodies in Churg-Strauss syndrome. Arthritis Rheum. 2005).
Thank you for your pointing out. We have changed based on your suggestion in abstract on page 1.
- In the chapter called “damage to each organ”, when you speak of severe asthma please discuss asthma severity at the diagnosis, which is linked to atopy (Berti A. et al. Severe/uncontrolled asthma and overall survival in atopic patients with eosinophilic granulomatosis with polyangitis. Rwspir Med. 2018; 142: 66-72) and long term asthma, which is linked with several factors (Berti A et al. Eosinophilic Granulomatosis with polyangitis clinical predictors of long-term asthma severity, Chest 2020).
Thank you for your valuable suggestions and we agree with your points. We have added and revised our manuscript based on your suggestion in damage to each organ section on page 2.
- I would change the structure of the paragraph, trying to divide biomarkers by area, i.e. biomarkers of asthma/eosinophilic disorders (e.g. IL5, Il13) and biomarkers of vasculitis, this last section is lacking. In this section authors should focus on those more inflammatory cytokines. I suggest to cite the paper of Pagnoux. C. et al. Serum cytokine and chemokine levels in patients with eosinophilic granulomatosis with polyangiitis, hypereosinophilic syndrome, or eosinophilic asthma Clin. Exp Rheum. 2019.
We divided the paragraphs into biomarkers for asthma and vasculitis.
- I would added a section on periostin and cite Rhee R et al. Serum periostin as a biomarker in eosinophilic granulomatosis with polyangitis. PLoS One. 2018 Oct 11: 13(10).
We have added periostin into biomarkers in the role of cytokines and chemokines section on page 5.
Reviewer 2 Report
The treatment of EGPA and in particular the role of anti-IL 5 monoclonal antibodies is a hot topic. Several publications have addressed this topic recently, and an increasing number of clinical data is now available. The role of the different cytokines in the pathophysiology of EGPA is sufficiently treated, however for a review, considering also the most recent publications, it is necessary to expand the discussion about the rationale and the clinical data relating to the use of mepolizumab for the treatment of EGPA.
Overall the manuscript is well conceived and structured, therefore, if properly integrated, it can be chosen for publication.
Author Response
- The treatment of EGPA and in particular the role of anti-IL-5 monoclonal antibodies is a hot topic. Several publications have addressed this topic recently, and an increasing number of clinical data is now available. The role of the different cytokines in the pathophysiology of EGPA is sufficiently treated, however for a review, considering also the most recent publications, it is necessary to expand the discussion about the rationale and clinical data relating to the use of mepolizumab for the treatment of EGPA.
Overall the manuscript is well conceived and structured, therefore, if properly integrated, it can be chosen for publication.
Thank you for pointing out. We have added new mepolizumab data in treatment of EGPA section on page 6 and 7.